# Development and Psychometric Characteristics of an Instrument to Assess Parental Feeding Practices to Promote Young Children’s Eating Self-Regulation: Results with a Portuguese Sample

**DOI:** 10.3390/nu14234953

**Published:** 2022-11-22

**Authors:** Ana Isabel Gomes, Magda Sofia Roberto, Ana Isabel Pereira, Cátia Alves, Patrícia João, Ana Rita Dias, João Veríssimo, Luísa Barros

**Affiliations:** 1Research Center for Psychological Science (CICPSI), Faculty of Psychology, University of Lisbon, Alameda da Universidade, 1649-013 Lisboa, Portugal; 2Faculty of Psychology, University of Lisbon, Alameda da Universidade, 1649-013 Lisboa, Portugal

**Keywords:** feeding practices, parents, preschool children, eating self-regulation, scale development, psychometric study

## Abstract

A parental child-centered feeding approach is likely to keep children’s biological mechanisms activated while eating, protecting them in an obesogenic context. However, few feeding practice measures assess parents’ behaviors to guide and prompt children to identify and respond appropriately to their signs of hunger and satiety. We aimed to develop and study the reliability, validity, and measurement invariance of a new scale to assess parental feeding practices to promote children’s self-regulation of food intake. To pursue this aim, we conducted two descriptive, cross-sectional, online studies in Portugal in an online format; a total of 536 parents of 2- to 6-year-old children completed the evaluation protocol. Factorial analysis findings support the theoretical organization proposed for the scale. The confirmatory factorial analysis supported a first-order factor structure with two subscales, *Prompting for eating self-regulation* and *Teaching about eating consequences*, with eight items in total. Both scales presented good internal consistency and adequate temporal stability, with a significant, positive, and moderate relationship. The results showed metric invariance for the child’s sex. Both types of practices were positively correlated with the child’s enjoyment of food. *Prompting for eating self-regulation* showed negative associations with parents’ emotional lack of control, children’s satiety responsiveness, slowness in eating, and fussiness. Preliminary studies confirmed both the validity and reliability of the instrument and the adequacy of adopting a self-regulatory approach when assessing child-centered feeding practices. Combining this instrument with others that assess coercive practices can be beneficial to capture ineffective parents’ behaviors on children’s eating self-regulation.

## 1. Introduction

Self-regulation of food intake refers to the ability to recognize and eat (or not eat) in response to internal feelings of hunger and fullness [1]. Two mechanisms are involved in the self-regulation of eating: *satiation*, i.e., the internal sensations that occur while eating and lead the individual to stop eating, and *satiety*, i.e., the physical signs that start after finishing eating and prevent the individual from eating again in the absence of hunger [2]. Healthy newborns can regulate food intake in response to hunger and satiety cues [3], and many preschool children maintain this capability [1]. Research has suggested that this eating self-regulation competence may be critical for children to maintain healthy eating patterns in a predominantly obesogenic environment [4,5] and has shown differences in this ability between children with normal weight and overweight [6,7].

Although children’s innate mechanisms of eating self-regulation can be protective, external aspects should be considered as potential contributors for individual variance in this capability during early childhood, such as parental–child interactions during feeding contexts [8]. For instance, variances in young children’s eating self-regulation have been linked to parental feeding practices [9,10]. Parenting practices are the actions parents engage in to influence their child’s corresponding behavior in a specific context, such as eating [11]. Most findings originated from studies that explored the associations between children’s eating self-regulation behaviors and parental coercive control practices. Pressure to eat or restriction of unhealthy foods and beverages can hinder young children’s natural ability to self-regulate their energy intake by providing external cues for eating instead of respecting children’s internal signs of hunger and fullness [12,13]. There is also substantial evidence that offering a structured feeding context, where parents provide consistent mealtime routines, define which foods are available to the child, and guide the child in making healthy food choices, can support children’s self-regulation of food intake and improve their healthy eating patterns [14,15]. Less is known about the role other responsive practices promoting a child’s autonomy in eating might have on the child’s capability. The lack of findings may be due to most feeding practices questionnaires being more focused on ineffective practices and rarely evaluating, for instance, parent’s identification and responses to the child’s signs of hunger and satiety or the child’s coaching to recognize these cues and respond appropriately, or to follow these internal sensations more than environmental prompts to eat [16,17]. This is a significant gap in further efforts to understand which aspects of mealtime parent–child interaction may determine children’s self-regulation in eating and to properly assess the impact of parenting interventions that promote responsive eating practices.

The present study emerged from the need for an instrument to assess responsive feeding practices, namely parental acknowledgment and responsiveness to children’s hunger and fullness, to study the efficacy of an intervention to change parents’ practices [18]. The *SmartFeeding4Kids* is an online, self-guided intervention to promote positive parental feeding practices and improve 2–6-year-old children’s diet, specifically, to increase vegetables, fruit, and legume intake and decrease intake of sugar-sweetened foods and beverages. The program, with seven sessions, uses self-regulatory behavior change techniques (e.g., self-monitoring, feedback, and goal setting) directed both at parents’ mealtime interaction with the child and the organization of the child’s feeding context.

Besides evaluating parental practices to prompt the child to recognize their internal cues both while eating and when the child asks for food outside meals, we also wanted to assess teaching practices about the consequences of food intake. Although children’s understanding of illness undergoes significant developmental changes during the first years of life [19], young children rely mostly on their direct perception to explain physical illness or discomfort and focus on people, objects, or external activities that are spatially or temporally close to the child or the sickness episodes, respectively [20]. However, they can learn simple explanations targeted to their cognitive and perceptual developmental level. Myand and Williams (2005) highlighted the parental influence on children’s concepts and beliefs by giving them advice related to health. Moreover, these authors showed that even young children could have some understanding of prevention when parents and schools offer sufficient education [21]. Providing information to children about the consequences of overeating, or overeating a specific food, can help them make healthier decisions. No single available and validated feeding practices instrument gave, in our opinion, sufficient attention to these specific responsive practices.

As such, the present study describes the development and validation of a scale to assess prompting and teaching practices to promote young children’s eating self-regulation, created to evaluate the efficacy of the *SmartFeeding4Kids* program. The examination of the instrument’s psychometric properties included the inspection of its factorial structure (both exploratory and confirmatory) and internal and test–retest consistency. The convergent and discriminant validity was examined through correlational analysis with the child’s eating behavior dimensions more or less related to satiation and satiety constructs (e.g., enjoyment of food and satiety responsiveness), parental concerns about the child’s weight, general parental emotional self-regulation, and children’s food intake of healthy and unhealthy foods. We also proposed to examine measurement invariance across sex as differences were found in young girls’ and boys’ eating behavior and their ability to self-regulate [6,22].

## 2. Materials and Methods

### 2.1. General Procedures

This study was conducted according to the guidelines in the Declaration of Helsinki, and all procedures involving research study participants were approved by the Ethics and Deontology Committee of the Faculty of Psychology, University of Lisbon, and the school boards involved. Written informed consent was obtained from all subjects/patients. The development and preliminary validation of the *Children’s Intake Self-Regulation Feeding Practices Scale* occurred in several stages. First, the construction of the instrument was based on earlier literature about parental feeding practices’ measurement and contributions from a panel of specialists in the area (see Section 2.2). Second, the scale was administered to parents of young children in two studies (see 3. Study 1 and 4. Study 2) to further develop and refine the scale and evaluate the scale’s reliability and validity. Data were collected between 8 January and 29 April 2019 (*Study 1*), 6 January and 2 March 2020, and 2 February and 9 June 2021 (*Study 2*).

Inclusion criteria included being a parent of a two- to a six-year-old child, cohabiting with the child, and understanding the Portuguese language. Considering that 251,060 Portuguese children between three and six years old were enrolled in kindergartens in 2021 [23], we estimated that the sample size needed for this study is approximately 248 participants, for a 99% confidence level and a 1% margin of error.

Parents were contacted to participate in this study through parent-directed social networks, daycare centers, or personal invitations addressed by email. According to school preferences, the evaluation protocol was available online through a Qualtrics platform link or in paper and pencil format; in the last case, additional instructions were given to ensure data confidentiality (e.g., delivery of the filled protocol in a sealed envelope). All parents read the consent form and agreed to participate in the study before accessing the protocol. To allow convergent/discriminant validity analysis of the scale, we asked a subgroup of parents to complete measures on the child’s dietary intake and eating behavior, concerns about the child’s weight, and parental emotion regulation. In another subsample, parents completed the parental feeding practices scale a second time, one month after the first administration, to assess test–retest reliability.

### 2.2. Development of the Scale

The scale development started with reviewing the most used instruments to assess parental feeding practices. We identified two instruments, the *Comprehensive Feeding Practices Questionnaire* [24] and the *Vegetable Parenting Practices Questionnaire* [25,26], that included some items about parental responsiveness to the child’s internal signs of hunger and fullness. We also considered several reviews of food parenting practices studies [4,14,27,28,29]. This process resulted in the generation/adaptation of nine items, organized into two main theoretical dimensions: *Teaching about eating consequences* (parent’s teaching about the consequences of eating more or less healthy and unhealthy foods) and *Prompting for eating self-regulation* (encouraging and helping the child to identify hunger and satiety internal cues). Then, we requested an analysis from two specialists (a nutritionist and a psychologist, both with experience and research in the area of child feeding) about the level of relevance, specificity, and clarity of each item, in a five-point Likert scale (ranging from 1, *Nothing* to 5, *Very much*). Finally, the items were tested with a group of six parents to assess the clarity of the items.

## 3. Study 1: Descriptive Analysis of the Items and Factorial Structure of the Scale

### 3.1. Sample Characteristics

The final sample consisted of 302 parents between 20 and 50 years old (M = 36.75; SD = 5.44). Most participants were mothers (87.4%) of one or two children (89.7%) and had a higher education degree (79.5%). Children were 2 to 6 years old (M = 3.78; SD = 1.30).

### 3.2. Measures

*Sociodemographic questionnaire:* We collected data about the parents’ age, sex, level of education, kinship with the child and number of children and adults in the household, and the child’s age.

*Children’s Intake Self-Regulation Feeding Practices Scale*: The nine-item preliminary version of the scale was answered on a five-point Likert scale (Totally false to Totally true), with higher values indicating more frequent use of each feeding practice.

### 3.3. Statistical Analysis

We began by performing a descriptive analysis of the initial nine-item scale to assess the percentage of participants that answered each response alternative and the skewness and kurtosis (statistic and standard error) values of each item. The distribution was considered approximately normal when the statistic/standard error was <2 for skewness and <7 for kurtosis. Next, we performed an exploratory factorial analysis of the scale. The sampling adequacy was verified with Bartlett’s sphericity tests (*p*-value inferior to 0.05) and the Kaiser–Meyer–Olkin measure (above 0.7). We used the principal components method and the Varimax rotation to examine the scale’s factor structure. The number of factors was identified by combining Horn’s parallel analysis [30], the scree plot, and the eigenvalue. We retained all items with factor loadings higher than 0.30 and differences between loadings in two or more factors superior to 0.10.

### 3.4. Results

#### 3.4.1. Descriptive Analysis of the Items

Some items showed higher asymmetry and/or kurtosis in both theoretical dimensions (Table 1). Considering that these findings may be related to the nature of the variable distribution in the Portuguese population, all items were retained for further analysis.

#### 3.4.2. Exploratory Factorial Analysis (EFA)

Parallel analysis, the scree plot analysis, and eigenvalues higher than 1 (Figure 1) suggested the retention of two components. One item did not comply with the defined criteria regarding factor loadings. Bartlett’s tests of sphericity (χ2 = 740.28, *p* < 0.001) and the Kaiser–Meyer–Olkin (KMO = 0.78) measure at the final analysis supported the sample adequacy. The EFA confirmed the theoretical dimensions of the scale and the two-factor structure explained 61.70% of the total variance (Table 2).

## 4. Study 2: Confirmation of the Factorial Structure, Measurement Invariance, Reliability, and Convergent/Discriminant Validity of the Scale

### 4.1. Sample Characteristics

Study 2 participants were 234 parents of young children between 19 and 58 years old (M = 36.74; SD = 5.72), primarily mothers (85.9%) with a higher education degree (60.7%). Their children were 2–6 years old (M = 4.32; SD = 1.31), 56.4% were male, 6.8% had a chronic health condition, and 3.4% received professional support owing to weight or eating problems. Most children lived with both parents (83.8%) and one or more siblings (56.0%); 52.6% of the families received child benefits.

### 4.2. Measures

*Sociodemographic questionnaire:* We collected general information about parents (age and sex, educational level, kinship with the child, household composition, and if parents received child benefits) and children (age and sex, childcare attendance, current professional support owing to weight or eating problems, and the existence of a chronic health condition).

*Children’s Intake Self-Regulation Feeding Practices Scale*: We adopted the eight-item scale structure suggested by the EFA for this study.

*Child Eating Behavior Questionnaire—CEBQ* [31]: The CEBQ is a 35-item instrument that evaluates children’s eating behavior and style through eight scales (i.e., Responsiveness to food, Enjoyment of food, Satiety responsiveness, Slowness in eating, Fussiness, Emotional overeating, Emotional undereating, and Desire for drinks). Parents answered each item on a five-point Likert scale according to the frequency of the behavior in their children (Never, Seldom, Sometimes, Often, Always). In this study, we used the Portuguese version of the instrument [32]. We found acceptable to good scale reliability coefficients (alphas between 0.675 for Emotional undereating and 0.895 for Desire for drinks).

*Parents’ concerns about child weight (subscale of the Child Feeding Questionnaire-Revised)* [33]: This three-item scale assesses parents’ concerns about the child’s risk of overweight (e.g., “How concerned are you about your child becoming overweight?”). The items were answered on a five-point Likert scale (from No concern to Very concerned); higher values correspond to higher concerns about the child’s weight. The scale showed good internal consistency (α = 0.814), similar to the results found in the Portuguese adaptation of the instrument [34].

*Parent Emotion Regulation Scale—PERS* [35]: The PERS is a 20-item scale that evaluates how parents respond to their child’s negative emotions. The instrument integrates four scales that access distinct dimensions of parental emotion regulation (i.e., Orientation to the child’s emotions, Avoidance of the child’s emotions, Emotional lack of control, and Acceptance of the child and parent’s emotions). The scale is answered on a five-point Likert scale (from Never or almost never to Always or almost always; scored 0–4, respectively). The reliability coefficients found were similar to those reported in the original study (acceptable to good Cronbach alpha’s between 0.615 for Emotional lack of control and 0.807 for Avoidance of child’s emotions), except for Acceptance of child and parent’s emotions (α = 0.449); as such, this subscale was excluded from further analysis.

*The Child’s Eating Habits Questionnaire—CEHQ* [36]: The CEHQ is a child’s food frequency questionnaire based on parents’ report of children’s dietary intake of healthy and unhealthy foods and beverages. In this study, parents were asked to indicate how frequently their child ate seven foods (i.e., vegetables, legumes, soup, fruit, desserts, candies/treats, and sodas) during the week on an eight-point scale (Never to Every day). Based on factorial and reliability analysis, the items were organized into two dimensions (Vegetables’ and fruits’ intake and Sugar-sweetened foods’ and beverages’ intake) with acceptable internal consistency (alphas between 0.59 and 0.57, respectively), considering the variability of foods accessed and the small number of items [37]. Higher mean values correspond to a more frequent intake of those foods.

### 4.3. Statistical Analysis

We performed a confirmatory factor analysis (CFA) with the maximum likelihood estimator (ML) to evaluate the adjustment of the structure extracted from the EFA. Specifically, we compared the derived first-order structure model (Model B) with a unifactorial model (Model A) to determine which solution best fits the data. We used the following fit indices to evaluate models’ adjustment: the chi-squared test (χ2), the comparative fit index (CFI), the Tucker–Lewis index (TLI), the standardized root mean square residual (SRMR), the root mean square error of approximation (RMSEA) with a 90% confidence interval, Akaike information criteria (AIC), and Bayesian information criteria (BIC). Evidence of adequate model fit occurred when CFI and TLI values were around 0.90 or greater [38,39], SRMR and RMSEA values below 0.08 [40,41], and smaller AIC and BIC values when models were compared [40,42]. Model comparison was also performed using the chi-square difference test [43]. To evaluate whether the factor loadings of the structural model depicted by the CFA are conceptually similar across gender, we performed multigroup measurement invariance by comparing the configural model with a metric one. Specifically, we compared the CFA factor structure to each gender group, allowing factor loadings to vary freely. Next, we constrained the factor loading structure, forcing it to be equivalent across groups. Finally, we compared both models’ fit and chi-square difference. Minor fit changes and non-significant chi-square differences suggest that item loadings perform similarly across gender groups, suggesting factor variances and covariances do not result from gender-based differences [44]. Analyses were performed using the following R [45] packages: lavaan [46] and semPlot [47].

After confirming the final factor structure of the scale, we run additional analysis to assess the reliability (internal consistency and test–retest) and convergent/discriminant validity of the scale. The internal consistency of the scale was evaluated with McDonald’s coefficient omega (ϖ) [48] and the inter-item correlation means (IICM), considering the total sample (*N* = 234). Internal consistency was considered good if omega values exceeded 0.70 and IICM values were higher than 0.20. Scale temporal stability was conducted with a subsample of 45 parents, considering a 30-day interval between administrations, and was measured with both Pearson and intra-class (ICC) correlation coefficients. We considered test–retest Pearson correlation values above 0.70 acceptable [49]. ICC was calculated based on an absolute-agreement, two-way mixed-effects model (single measures); we considered ICC values between 0.4 and 0.75 as good and above 0.75 as excellent [50]. The convergent/discriminant validity was accessed through correlational analysis between *Children’s Intake Self-Regulation Feeding Practices Scale*, parental concerns about child’s weight, child’s eating behavior, child’s intake of vegetables/fruits and sugar-sweetened foods and beverages, and parental emotional regulation, considering a subsample of 132 parents. Statistical significance of the tests was achieved for *p* < 0.05.

### 4.4. Results

#### 4.4.1. Confirmatory Factorial Analysis and Measurement Invariance

The CFA results revealed that Model B was the factorial solution with better adjustment (Table 3). All factor loadings were significant (*p <* 0.001). Figure 2 depicts the first-order factor structure of the *Children’s Intake Self-Regulation Feeding Practices Scale*. Table 3 also provides the factorial adjustment for the multigroup invariance model. The results provided evidence supporting metric invariance according to gender.

#### 4.4.2. Internal Consistency Reliability and Correlation between Subscales

Both scales showed good internal consistency coefficients (*Teaching about eating consequences*: ϖ = 0.72; *Prompting for eating self-regulation*: ϖ = 0.71). IICM values were also adequate (0.37 for *Prompting for eating self-regulation* and 0.42 for *Teaching about eating consequences*). The correlation between the scales was significant, positive, and moderate (r_s_ = 0.308, *p* < 0.01).

#### 4.4.3. Test–Retest Reliability

We found moderate to high significant correlations between test–retest (0.50 for *Teaching about eating consequences* and 0.71 for *Prompting for eating self-regulation*), although the results for the first factor scored below the cut-off values suggested by Nunnally and Bernstein [49]. The ICC values of the scales were good (0.49 for *Teaching about eating consequences* and 0.71 for *Prompting for eating self-regulation*).

#### 4.4.4. Convergent and Discriminant Analysis

We found several significant weak correlations between the *Children’s Intake Self-Regulation Feeding Practices* scales and the child’s and parent’s dimensions (Table 4). Higher scores on *Teaching about eating consequences* and *Prompting for eating self-regulation* practices were associated with higher scores on *Enjoyment of Food*. Moreover, parents that reported prompting their child more frequently towards their internal sensations of hunger and fullness scored lower on emotional lack of control strategies and had children with less satiety responsiveness, less slowness in eating, and fewer fussiness behaviors.

#### 4.4.5. Descriptive Analysis of the Sample

The scale scores were calculated according to the proposed CFA structure through the item answers’ mean, achieving a possible score range of 1 to 5 in each scale. Parents reported moderate (*Prompting for eating self-regulation*: M = 3.39, SD = 0.68) to frequent (*Teaching about eating consequences*: M = 4.24, SD = 0.54) use of the two types of feeding practices to promote children’s self-regulation of food intake.

## 5. Discussion

Parental practices that help children decide about starting or continuing to eat based on identifying internal cues of hunger and fullness have been less studied as an independent dimension and are less represented in existing parental feeding practices instruments [16]. This should include the caregivers’ practices that support the children identifying and signaling hunger or satiety and adults responding appropriately. The most robust measures of responsive feeding practices do not include this dimension (Responsiveness to cues/Child autonomy) [17]. Thus, within the scope of the *SmartFeeding4Kids* program, we proposed to design an instrument to evaluate these practices and examine the scale’s validity, reliability, and measurement invariance regarding children’s sex.

The psychometric findings support the theoretical organization proposed in developing the scale. CFA indicated a first-order structure with two factors, *Teaching about eating consequences* and *Prompting for eating self-regulation*, as suitable for this scale, with the eight items equally distributed by those dimensions. The adequacy of the final structure is also confirmed by the two scales’ good internal consistency and IICM. Additionally, the correlation between the scales was moderate, which supports our proposal of two related but distinctive types of feeding practices to promote children’s self-regulation of food intake.

Test–retest and measurement invariance has rarely been assessed in earlier validation studies of parental feeding practices questionnaires [16,27]. In our scale, the temporal stability was not confirmed for both scales when adopting the cut-offs proposed by Nunnally and Bernstein [49] for Pearson coefficient analysis, which can be explained by some expected variation in the use of different feeding practices with young children in a short period; however, the ICC values validated the stability of the instrument. Because of the sample size by group, the number of variables included in the analysis and the number of multigroup models tested were restricted to the child’s sex and metric invariance, respectively, to avoid decreasing the power of the performed tests. Further studies should continue to perform additional invariance analysis (specifically scalar invariance) regarding children’s sex and expand this assessment considering other relevant children and parents’ variables (e.g., children’s age and family socioeconomic status). Extending the recruitment to parents of older children will also allow understanding of whether the instrument remains adequate to assess practices that promote child self-regulation at other stages of development, considering that how parents influence the child’s eating behavior can change in response to a greater child autonomy regarding food choices [15,27].

We also assessed the convergent and discriminant validity of the instrument. We found that both parental feeding practices showed significant positive associations with children’s enjoyment of food. On the other hand, the *Prompting for eating self-regulation* scale showed negative associations with three dimensions of the child’s eating behavior, i.e., satiety responsiveness, slowness in eating, and fussiness. A recently published instrument (i.e., Food Parenting Inventory, FPI) that included a specific subscale to evaluate parental responsiveness to the child’s fullness cues did not find significant correlations with any CEBQ subscale [51]. Specific features of the sample (low-income Latina parents with a lower education level were considered for the FPI validation study) may be responsible for the differences between studies. Although conclusions must be cautiously made owing to the weak correlations found, our findings might indicate that parents, based on different children’s eating behaviors (food approach or avoidance), implement prompting or teaching strategies to promote the child’s eating self-regulation. A child’s more intense expression of pleasure while eating can be a warning sign of food dysregulation and lead parents to mobilize more strategies to support child’s identification of internal cues of fullness and to alert for the consequences of excessive eating. On the other hand, parents can find it more beneficial to focus only on the child’s identification of their internal sensations if the child eats faster (i.e., slowness in eating) or attend less to their signs (e.g., satiety responsiveness).

It is noticeable that *Prompting for eating self-regulation* was negatively correlated with the parents’ difficulties in managing their own emotions when interacting with the child. Similar relationships were found in a sample of Portuguese preschool children’s parents [52]. These findings corroborate the link Frankel and colleagues (2012) and Russel and Russel [5] suggested between the general mechanisms parents use to manage their own emotions and their children’s emotions and the type of strategies they use to maintain or promote children’s self-regulation of food intake.

This study has some limitations. The participants were mainly from the Lisbon region, with mostly mothers with a post-secondary degree, limiting the generalizability of the results. All data analyzed were reported by parents, which can increase the impact of the social desirability and bias on findings; further studies may consider the inclusion of observational measures to complement the validity studies of this instrument. Moreover, the second study was conducted during the COVID-19 pandemic lockdown. Earlier studies [53,54] had concluded that the different COVID-19 lockdowns significantly impacted eating behaviors within families.

## 6. Conclusions

Preliminary studies confirmed the validity and reliability of the *Children’s Intake Self-Regulation Feeding Practices Scale,* reinforcing the pertinence of adopting a self-regulatory perspective when studying parenting child-centered feeding practices. A child-centered feeding approach is likely to respect the child’s behaviors driven by internal biological cues or sensations when eating [17] and is positively related to children’s fruit and vegetable intake [55]. This measure is a novel contribution and can be used to assess parenting feeding practices to regulate the child’s eating behavior and in intervention studies focused on promoting effective parenting practices. Moreover, this scale should be combined with instruments that assess coercive control and restrictive practices. Those practices can be viewed as opposed to those assessed by our measure and have been shown to affect the children’s self-regulation negatively. Nevertheless, additional studies are needed to further examine the instrument’s validity, focusing on greater diversity in the sample of parents and children, and expanding the study of measurement invariance.

## Figures and Tables

**Figure 1 nutrients-14-04953-f001:**
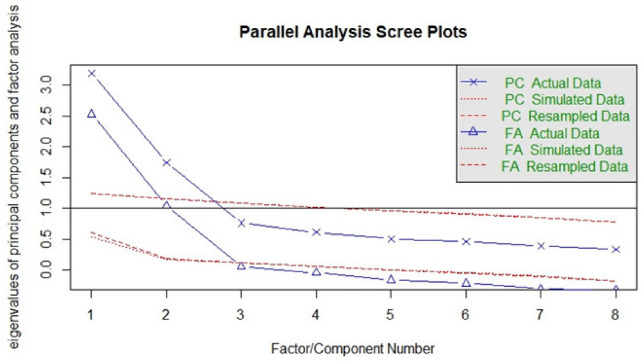
Parallel analysis (N = 302).

**Figure 2 nutrients-14-04953-f002:**
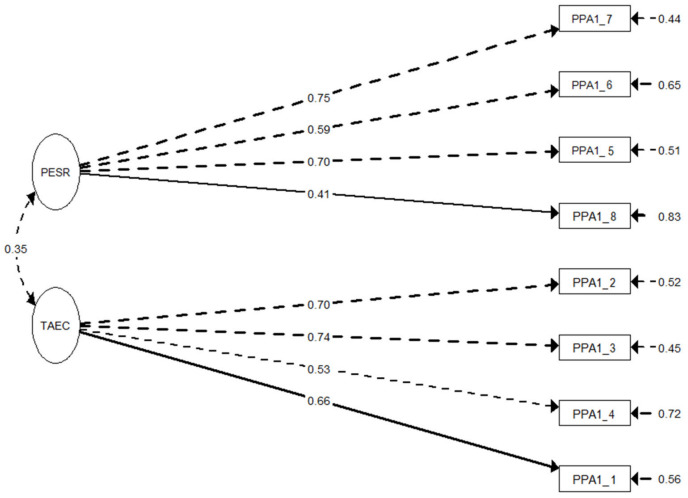
First-order factor structure, standardized factor loadings, and variances of the Children’s Intake Self-Regulation Feeding Practices Scale. Note. Latent factor labels reflect the following dimensions: PESR (Prompting of eating self-regulation) and TAEC (Teaching about eating consequences). Solid lines illustrate the item whose factor loading was constrained to 1.

**Table 1 nutrients-14-04953-t001:** Descriptive analysis of the initial items (N = 302).

Items	Totally False (%)	False (%)	Neither True nor False (%)	True (%)	Totally True (%)	Skewness (Std. Error)	Kurtosis (Std. Error)
I tell my child what will happen (e.g., upset stomach, feeling sick) if she eats too many unhealthy foods.	1.3	2.6	6.6	45.0	44.4	−1.470 (0.140)	3.063 (0.280)
I explain to the child that we can get sick or feel less energy to play when we eat a lot.	1.7	7.9	16.9	44.7	28.8	−0.822 (0.140)	0.326 (0.280)
I tell my child that eating vegetables and fruits is important, because they make us feel good and with energy.	0.7	1.0	7.6	50.0	40.7	−1.107 (0.140)	2.586 (0.280)
I explain to the child that some foods, like treats and desserts, should only be eaten occasionally and in small amounts.	1.0	2.0	8.9	44.4	43.7	−1.127 (0.140)	2.407 (0.280)
I serve small amounts of food to the child and let her have more if she wants.	0.3	2.3	14.2	61.3	21.9	−0.681 (0.140)	1.401 (0.280)
If the child says she wants seconds, but I think she had enough, I encourage her to stop eating.	3.0	20.5	22.2	44.7	9.6	−0.413 (0.140)	−0.642 (0.280)
When the child says she wants seconds, I ask her if she has an appetite or if she wants more just because she likes that food a lot.	4.0	17.9	32.8	35.4	9.9	−0.260 (0.140)	−0.466 (0.280)
When the child asks for food outside meal times, I ask her to think if she is hungry or just feeling bored.	7.3	33.8	37.7	17.2	4.0	0.247 (0.140)	−0.292 (0.280)
I try to help the child identify when she is satisfied or still has an appetite.	4.3	12.9	33.8	39.7	9.3	−0.461 (0.140)	−0.081 (0.280)

**Table 2 nutrients-14-04953-t002:** Exploratory factorial analysis of the instrument (N = 302).

Items	Factor 1. *Teaching about Eating Consequences*	Factor 2. *Prompting of Eating Self-Regulation*
1. I tell my child what will happen (e.g., upset stomach, feeling sick) if she eats too many unhealthy foods.	0.809	
2. I explain to the child that some foods, like treats and desserts, should only be eaten occasionally and in small amounts.	0.805	
3. I tell my child eating vegetables and fruits is important, because they make us feel good and with energy.	0.797	
4. I explain to the child that we can get sick or feel less energy to play when we eat a lot.	0.733	
5. When the child says she wants seconds, I ask her if she has an appetite or if she wants more just because she likes that food a lot.		0.836
6. When the child asks for food outside meal times, I ask her to think if she is hungry or just feeling bored.		0.752
7. I try to help the child identify when she is satisfied or still has an appetite.		0.730
8. If the child says she wants seconds, but I think she had enough, I encourage her to stop eating.		0.705
** *% Explained variance* **	31.77	29.93
** *Eigenvalue* **	2.54	2.40

**Table 3 nutrients-14-04953-t003:** Goodness-of-fit statistics for the CFA and multigroup invariance models (N = 234).

	χ^2^	*df*	CFI	TLI	SRMR	RMSEA	90% CI RMSEA	AIC	BIC	*df*, Δχ^2^	Model Comparison
CFA Models											
- Model A	177.705 ***	20	0.63	0.48	0.12	0.18	[0.16, 0.21]	4548.508	4603.793		_
- Model B	58.656 ***	19	0.91	0.86	0.08	0.09	[0.07, 0.12]	4431.458	4490.198	1, 119.05 ***	Model A
Gender Multigroup Invariance											
- Configural Model	85.718 ***	38	0.89	0.84	0.09	0.10	[0.07, 0.13]	4464.896	4637.662		
- Metric Model	94.870 ***	44	0.89	0.86	0.09	0.10	[0.07, 0.13]	4462.048	4614.083	6, 9.1526	Configural

Note. χ^2^ = Chi-squared test; *df* = degrees of freedom; CFI = comparative fit index; TLI = Tucker–Lewis Index; SRMR = standardized root mean square residual; RMSEA = root mean square error of approximation; AIC = Akaike information criteria; BIC = Bayesian information criteria. Model A (unidimensional structure), Model B (two first-order factor structure). For multigroup invariance models, the sample size by gender was 132 boys and 102 girls. **** p* < 0.001.

**Table 4 nutrients-14-04953-t004:** Correlations between *Children’s Intake Self-Regulation Feeding Practices Questionnaire* and CEBQ dimensions, Concern of Child Weight subscale (CFQ-R), PERS (Orientation to child’s emotions, Avoidance of child’s emotions, Emotional lack of control), and CEHQ subscales (N = 132).

Variables	*Teaching about Eating Consequences*	*Prompting of Eating Self-Regulation*
***Responsiveness to food*** **(CEBQ)**	0.062	0.169
***Enjoyment of food*** **(CEBQ)**	0.245 **	0.277 **
***Satiety responsiveness*** **(CEBQ)**	−0.101	−0.277**
***Slowness in eating*** **(CEBQ)**	−0.102	−0.225 **
***Fussiness*** **(CEBQ)**	−0.045	−0.177 **
***Emotional overeating*** **(CEBQ)**	−0.091	0.054
***Emotional undereating*** **(CEBQ)**	0.018	0.008
***Desire for drinks*** **(CEBQ)**	−0.135	0.034
***Concern of child weight*** **(CFQ-R)**	0.156	0.137
***Orientation to child’s emotions*** **(PERS)**	0.125	0.123
***Avoidance of child’s emotions*** **(PERS)**	0.020	0.151
***Emotional lack of control*** **(PERS)**	−0.055	−0.179 *
***Vegetables’ and fruits’ intake*** **(CEHQ)**	0.115	0.099
***Sugar-sweetened foods’ and beverages’ intake*** **(CEHQ)**	0.094	0.033

Note. Spearman correlation coefficient. ** p* < 0.05, *** p* < 0.01.

## Data Availability

The data presented in this study are available on request from the corresponding author. The data are not publicly available due to Institutional Ethics Committee restrictions.

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
