# Peer review of "Development and Psychometric Characteristics of an Instrument to Assess Parental Feeding Practices to Promote Young Children’s Eating Self-Regulation: Results with a Portuguese Sample"

_nutrients, 2022, doi:10.3390/nu14234953_

Round 1
Reviewer 1 Report
Dear Authors,
I was pleased to review your manuscript on the development and psychometric characterization of an instrument for assessing parents' feeding practices to promote young children's nutritional self-regulation. It is a correctly designed study that needs only minor improvements.
1) Please calculate and provide a minimum sample size that would be representative of your study.
2. The inclusion criteria for the survey should be better described.
3. conducting surveys using the CAWI method does not avoid the common phenomenon of "bot/fakeresponders", which is characterized by surveys made available on the basis of online forms, how did you try to avoid this error?
4. it is also worth emphasizing the relevance of the tools presented for use in determining the prevalence of neophobic behavior in children.
Greetings
Author Response
Response to reviewers
Dear editor and reviewers,
We thank the Editor and Reviewers for taking the time to review our article and for all the relevant suggestions. We considered all the suggestions and included them throughout the manuscript. Below we answer all comments, indicating the changes made and providing additional explanations about the highlighted issues.
Reviewer 1
Question |
Answer |
1) Please calculate and provide a minimum sample size that would be representative of your study. |
We added a paragraph where the minimum sample size calculation is reported. Despite the sample collected in this study being sufficient, taking into account the minimal size calculation, the study design, and the statistical procedures carried out, it is noticeable the sample is not representative in geographical terms. |
2. The inclusion criteria for the survey should be better described. |
Thank you for noticing. We clarified the inclusion criteria for this study. |
3. Conducting surveys using the CAWI method does not avoid the common phenomenon of "bot/fakeresponders", which is characterized by surveys made available on the basis of online forms, how did you try to avoid this error? |
We aimed to control parents’ responses through the forums of dissemination. Participants were mainly gathered through childcare centers and kindergartens (where parents received an invitation letter), through groups of parents known to the research team, or through parent-directed social networks (such as young children’s parents’ Facebook pages, where we asked the moderator to disseminate the study). Qualtrics XM includes bot detection and prevents indexing. |
4. it is also worth emphasizing the relevance of the tools presented for use in determining the prevalence of neophobic behavior in children. |
The instrument presented is focused on assessing parental feeding behaviors that promote children’s self-regulation while eating. It was not developed to measure children’s specific eating behaviors, like neophobia. |

Reviewer 2 Report
1. The scientific article is properly structured.
2. The text contains new content. The problem is correctly presented.
3. This is the original text. The authors complete the "blank spot" in the results of scientific research.
4. The results and conclusions of the research will be of interest to the readers.
5. Remarks - Table 3 is poorly constructed. The data is hardly visible to the reader.
6. The summary needs correction. It's too short. The conclusions should be broadened, the results discussed more extensively and comparisons made with the results of other researchers.
Author Response
Response to reviewers
Dear editor and reviewers,
We thank the Editor and Reviewers for taking the time to review our article and for all the relevant suggestions. We considered all the suggestions and included them throughout the manuscript. Below we answer all comments, indicating the changes made and providing additional explanations about the highlighted issues.
Reviewer 2
Question |
Answer |
Remarks - Table 3 is poorly constructed. The data is hardly visible to the reader. |
Thank you for noticing that. Some tables were not appropriately formatted in this template. We revised all tables to guarantee that the data can be easily read. |
The summary needs correction. It's too short. The conclusions should be broadened, the results discussed more extensively and comparisons made with the results of other researchers. |
We revised the abstract and included some additional information regarding background and conclusions. Although we revised the discussion considering your comments and included some additional information, we found it hard to enhance the comparison with similar studies, considering that few instruments specifically evaluate this kind of parental feeding practices. |
